# Assessing Auditory Processing in Children with Listening Difficulties: A Pilot Study

**DOI:** 10.3390/jcm12030897

**Published:** 2023-01-23

**Authors:** Shaghayegh Omidvar, Fauve Duquette-Laplante, Caryn Bursch, Benoît Jutras, Amineh Koravand

**Affiliations:** 1Audiology and Speech Pathology Program, School of Rehabilitation Sciences, Faculty of Health Sciences, University of Ottawa, Ottawa, ON K1H 8L, Canada; 2School of Speech-Language Pathology and Audiology, Université de Montréal, Montreal, QC H3C 3J7, Canada; 3APD Ottawa, Ottawa, ON K2V 5G7, Canada; 4Research Centre, CHU Sainte-Justine, Montreal, QC H3T 1C5, Canada

**Keywords:** listening difficulties, auditory processing, children, frequency following response, auditory behavioural tests, auditory brainstem responses

## Abstract

Background: Auditory processing disorders (APD) may be one of the problems experienced by children with listening difficulties (LiD). The combination of auditory behavioural and electrophysiological tests could help to provide a better understanding of the abilities/disabilities of children with LiD. The current study aimed to quantify the auditory processing abilities and function in children with LiD. Methods: Twenty children, ten with LiD (age = 8.46; SD = 1.39) and ten typically developing (TD) (age = 9.45; SD = 1.57) participated in this study. All children were evaluated with auditory processing tests as well as with attention and phonemic synthesis tasks. Electrophysiological measures were also conducted with click and speech auditory brainstem responses (ABR). Results: Children with LiD performed significantly worse than TD children for most behavioural tasks, indicating shortcomings in functional auditory processing. Moreover, the click-ABR wave I amplitude was smaller, and the speech-ABR waves D and E latencies were longer for the LiD children compared to the results of TD children. No significant difference was found when evaluating neural correlates between groups. Conclusions: Combining behavioural testing with click-ABR and speech-ABR can highlight functional and neurophysiological deficiencies in children with learning and listening issues, especially at the brainstem level.

## 1. Introduction

Listening difficulties (LiD) can have a negative impact on academic learning as 60% of school time for children in the primary school level is devoted to listening, according to Imhof (2008) [1]. As suggested by Dawes and Bishop (2009) [2] and illustrated by Dillon et al., (2021) [3], LiD could be caused by hearing, auditory processing, language and/or cognitive deficits. The terms auditory processing deficits or disorders (APD) can be interchangeable, but APD and LiD are not commutable terms, since APD would be under the umbrella term of LiD [3].

Many APD definitions have been proposed [4,5,6,7,8], and the majority agree that APDs are failures of the central auditory system that lead to difficulties in processing auditory information. The potential coexistence of auditory processing, language, and cognitive deficits could be a challenge when assessing children suspected of APD, as no universal test battery yet exists to determine the underlying cause of their difficulties [9]. In addition, attention [10], motivation [11], and linguistic ability [12] can easily affect performance when using behavioural tests.

Accordingly, electrophysiological measurements could be utilized to reduce the contribution of attention, motivation and other non-auditory factors when evaluating auditory abilities in children. Since these measures do not rely on the listener’s participation, they can provide a better insight into understanding auditory system function [13]. The click-evoked ABR is the most common transient electrophysiological response method that can detect activation of the auditory pathway from the cochlea to the rostral brainstem following stimulus presentation [14]. Studies have primarily shown prolonged latency of click-evoked auditory brainstem response (ABR) in children with or at risk of APD, especially while using faster presentation rates (such as 57.7 presentations per second–pps) than slower ones (such as 13.3 pps) [15,16,17,18,19]. Significant reductions in amplitudes of wave V [16,20,21,22], wave III [16,21], and wave I [21] were also reported in APD children relative to control groups.

While click-like stimuli were utilized in the early investigations of ABR to optimize the transient auditory brainstem responses, they are not a good representation of relevant behavioural sounds heard in the real world, such as speech and music, non-speech vocal, and environmental sounds [23]. Additional methods were developed with speech stimuli that are spectrally and temporally more sophisticated than click stimuli and can provide more specific information about auditory processing [23,24,25]. For example, when a syllable such as /da/ is pronounced, there is a burst release of the consonant /d/, followed by a transition portion from the consonant to the vowel and the sustained portion of the vowel. The electrophysiological recording of the transient and sustained portions of speech stimuli at the midbrain level [23] represents the frequency-following response (FFR) [26]. In FFR recordings, the transient portion is marked by waves V and A, recorded 6 to 10 ms after the stimulus onset, followed by wave C representing the formant transition. Waves D, E, and F appearing later are associated with the periodicity of the vowel, and at the end, wave O represents the offset of stimulus [27]. These responses are influenced by cognitive, sensory, and reward inputs due to the intersection of afferent and efferent auditory projections in the midbrain, and non-auditory cortices [26]. The FFR was previously called the complex or speech-evoked ABR. However, because this term is unable to show the integrated and experience-dependent nature of the auditory neural activity, this was replaced by the term FFR [26]. The FFR components have both subcortical and cortical origins [28,29,30,31]. 

Previous studies using FFR reported significantly longer latencies of some transient and/or sustained elements in children with LiD relative to the controls [32,33,34,35,36,37]. Kumar and Singh (2015) reported significant latency prolongations of waves V and A and reductions in V/A slope in these children compared to the controls [32]. Similar results were obtained by Rocha-Muniz et al., (2012, 2014) [33,34]. Three groups of school-aged children were involved in that study: children with APD, children with language impairment (LI), and typically developing children (TD). Results showed that the children with APD and/or children with LI showed longer latency for wave A than the TD group. There was also a significant increase in the latency of wave V in the LI group compared with the TD [33,34] and APD [34] groups. Additionally, a substantial difference between the TD/APD [33] and TD/LI [33,34] groups relative to the waves C and O was observed, respectively. The latency of peaks D, E, and F did not differ significantly between the TD and APD groups [33,34]. There were significant differences in the latencies of waves D [34], E, and F [33,34] between the LI group and the TD/APD groups [34]. Gabr and Darwish (2016) also reported longer latencies and smaller amplitudes of all waves from V to O in children with specific language impairment (SLI) relative to the TD children. Similar results were obtained with a speech stimulus in a group of children with APD, where significantly longer latencies and smaller amplitudes were measured for waves V, A, C, and F compared to the controls [38]. The amplitude and latency of waves E and O were not analyzed in the Filippini and Schochat (2009) study [38], which discards potentially valuable information. Furthermore, they recruited children and adults for their study, and data analysis was provided without distinguishing the results of children from those of adults [38]. It is therefore difficult to know precisely the responses associated only with the electrophysiological measurements of the children. When using clicks instead of speech stimuli, Filippini and Schochat (2009) did not find a significant difference between the ABR components of children with APD and those of TD children [38]. This was also seen in a group of children with learning disabilities (LD) [35]. 

In summary, there is little literature on the auditory capacities and function of children with LiD in order to determine whether their difficulties might be caused primarily by APD. Moreover, none of the above studies on FFR investigated the neural consistency of recorded electrophysiological responses in children with LiD. 

The aim of the present study was to explore the auditory abilities and function of children with LiD and learning difficulties. Two sub-objectives were targeted: (1) identify whether their LiD was specifically related to APD and (2) examine whether the click-evoked ABR and FFR components could be sensitive to differences between responses of children with LiD and those of TD children.

## 2. Methods

The present study was approved by the Office of Research Ethics and Integrity at the University of Ottawa (the ethics committee approval code # H-03-18-427). The study was explained to the participants and legal guardians. Legal guardians were given a consent form to read, and all gave written consent for their child to participate in the current study. Children also provided their assent before the data collection. 

### 2.1. Participants

This cross-sectional comparative study was conducted at the University of Ottawa. Eleven experimental group participants (mean age: 8.46; SD: ±1.39, 3 girls, 8 boys) were recruited at a clinic that tests for APD. They were referred to the clinic because of their difficulties in school (see details in Table 1). One participant was excluded from the experimental data set as he was diagnosed with autism spectrum disorder. Six of the children had a learning disorder, and four had an ADHD diagnosis—two of them were medicated. Ten participants in the control group were recruited in the community (mean age: 9.45; SD: ±1.57, 8 girls, 2 boys). There was no significant difference between the experimental group of children and those of the control group regarding age (U = 40.00, *p* = 0.481). The control group participants resided in the same province as the participants of the experimental group. They had no known otologic, congenital, neurological, developmental or metabolic disorders. None of the participants had a diagnosis of intellectual disability. Participants in both groups had normal peripheral auditory function: otoscopy was unremarkable, tympanometry values were between 0.5 and 2.5 mL for ear canal volume; middle ear pressure was between −150 to 50 daPa; tympanic membrane compliance was between 0.3 to 2 mmho, and conventional pure tone audiometry revealed hearing thresholds equal to or below 20 dB HL for all frequencies tested (250 to 8000 Hz). All participants spoke English fluently.

### 2.2. Material and Procedure

The evaluation comprised two types of assessment: (1) auditory processing evaluation and (2) electrophysiological evaluation. The auditory processing tests and the electrophysiological recordings were administered to the participants at the APD clinic or at the University of Ottawa. All the evaluations occurred in a soundproof booth, respecting standard noise floor levels (ANSI S3.1-1999, R2008) [39].

#### 2.2.1. Auditory Processing Evaluation 

Participants underwent a battery of behavioural auditory processing tests. All children performed at least one test of each of the following auditory ability categories: binaural integration, binaural separation, figure-ground separation, sequential organization with variations in frequency, temporal resolution, and auditory closure. Two complementary tests were included in this battery, one assessing phonemic synthesis ability and one evaluating sustained auditory attention. During the testing, multiple breaks were given as needed by the children. 

##### Binaural Integration

The Dichotic Digit test (DD) [40] or the Staggered Spondaic Word test (SSW) [41] was used to assess binaural integration ability at a level of 50 dB SL relative to the pure tone average. For the DD test, participants were asked to repeat 20 sets of four different numbers with pairs of two numbers in each ear at the same time. The percentage of correct responses was calculated for each ear. For the SSW, they heard 40 sets of four words presented in this specific sequence: one single word in one ear, two different words in both ears at the same time and the last word in the other ear. The participants repeated the four words in the order they were presented. The number of errors was calculated for the four listening conditions.

##### Binaural Separation

Binaural separation ability was evaluated with the Competing Sentences Test (CST) [42]. The participants heard 20 sets of two different sentences simultaneously, one in each ear. They were asked to pay attention to and repeat the sentence presented in one ear at 35 dB HL—the target ear—and ignore the sentence presented at 50 dB HL in the opposite ear—the non-target ear. The score corresponded to the percentage of correct responses in each ear.

##### Figure-Ground Segregation

The Bamford–Kowal–Bench Speech in Noise test (BKBSIN) [43,44] was utilized to test figure-ground segregation ability. Sentences embedded in babble noise were presented at 50 dB HL to each ear and binaurally. Each sentence was preceded by the word “Ready” so that the participant paid attention to the sentence. As the background voices became progressively louder, the sentences became harder to hear and repeat. The participants were instructed to ignore the background talkers and repeat the sentences after each presentation. As recommended by Bench et al., (1979) [42], the test was scored by adding the correct number of words recalled and subtracting each total from 23.5. Two lists were averaged for each condition: binaural presentation, right ear and left ear separately. Then age corrections were applied to calculate the signal-to-noise ratio loss for each condition [43].

##### Auditory Closure

Auditory closure ability was evaluated with the Low Pass Filtered Speech test (LPFST) [45,46], where single words were low-pass filtered at 1000 Hz and presented at a level of 50 dB HL. Each ear was tested individually with a list of 25 words, and participants had to repeat each word. The percentage of correctly repeated words was determined for each ear.

##### Auditory Sequential Organization 

Auditory sequential organization was evaluated with the Pitch Pattern Sequence Test (PPST) [47], where participants heard 20 sets of three sounds presented monaurally at 50 dB SL relative to the auditory threshold at 1000 Hz. Each pattern was made with a combination of two different frequencies: high (1122 Hz) or low (880 Hz). The participants listened to each group and correctly hummed or labelled each sound in the correct order. The score was the percentage of correct responses for each ear.

##### Auditory Temporal Resolution

The Random Gap Detection Test (RGDT) [48] assessed auditory temporal resolution. A single sound or pairs of sounds were presented binaurally to the participants at a level of 55 dB HL. Pairs of sounds were separated by an interval of silence varying randomly from 2 to 40 msec. The participants were asked to identify if they heard one or two sounds. The auditory temporal resolution threshold corresponded to the smallest silence interval for which participants consistently identified two sounds instead of one.

##### Phonemic Synthesis and Sustained Attention

Two complimentary evaluations were conducted: (1) The Phonemic Synthesis (PS) test [49], in which sequences of phonemes were presented binaurally at 50 dB HL. The participant’s task was to combine these phonemes to identify the words they made. The test was scored by adding all the correct answers. (2) The Auditory Continuous Performance Test (ACPT), in which participants were instructed to press a button only, and every time they heard the word dog in an 11-min-long word series [50]. The words were presented at 50 dB HL or at the most comfortable level identified by the participants.

#### 2.2.2. Electrophysiological Evaluation 

Throughout the electrophysiological testing, participants were seated comfortably watching a muted movie with subtitles in a dimly lit room. They were instructed to remain relaxed and to refrain from moving or speaking. Both click- and speech-ABR took approximately 40 min, and breaks were given as needed. Two stimuli were generated by BioMARK^®^ (Biological Marker of Auditory Processing, BioMARK software, NavigatorPro AEP system, Bio-logic Systems Corp., Orlando, FL, USA): (1) a click of 100 μsec broad-spectrum square wave and (2) a five-formant 40 msec /da/, with an initial noise burst and a formant transition between the vowel and the consonant. They were presented at 80 dB SPL in the right ear only as there is evidence of a speech-specific right ear advantage [51] through an insert earphone (EARLINK 3B,3M Auditory Systems, Indianapolis, IN, USA). A double-channel montage was used with four disposable adhesive scalp electrodes (Natus Medical Inc., Mundelein, IL, USA). According to the Jasper (1958) 10–20 system, four electrodes were placed on the scalp: a non-invertive electrode in Cz (vertex), two inverted electrodes at the mastoids or earlobes, and a grounded electrode on the forehead [52]. Electrical impedance for each electrode was not greater than 5 kΩ, and discrepancies between electrodes did not exceed 2 kΩ. Each participant completed two blocks averaging 1500 artifact-free sweeps per recording for the click-ABR recordings and two artifact-free blocks with 2000 sweeps per recording for the speech-ABR recordings. When averaging, trials with artifacts exceeding ±23.8 µV were rejected; the total number of artifacts did not surpass 10% of the total number of sweeps. The clicks were presented at a rarefaction polarity with a rate of 13.3 clicks/sec and a 10.6 ms time window. The recordings were filtered online using an 80 to 1500 Hz band-pass with a 12 dB/octave filter roll-off. The /da/ syllable was recorded using an 85.33 ms and 15-ms pre-stimulus time window with a presentation rate of 10.9 cycles/sec and an alternating polarity. The data were filtered online from 100 to 2000 Hz. 

Results for the electrophysiological measures were computed in four ways: (1) transient responses for click-ABR and speech-ABR, (2) sustained responses, (3) neural response consistency, and (4) the stimulus-to-response consistency for speech-ABRs.

##### Transient Response

Peak selection was completed through the BioMARK software (NavigatorPro AEP system, Bio-logic Systems Corp). For click-ABR, three peaks—I, III, and V—were identified and marked manually. For speech-ABR, seven peaks were identified and marked manually: two onsets—V and A, a consonant-vowel transition C, three steady states—D, E, F—and an offset—O. Two evaluators identified all peaks on each recording separately, and the inter-judge concordance was 96% for the click-ABR and 98% for the speech-ABR. If both evaluators could not agree on a peak, a third evaluator would break the tie. Following the peak selection, electrophysiological data were extracted with the MATLAB [53] powered Brainstem Toolbox Version 2013 [54].

##### Sustained Response

To calculate the sustained responses of the speech-ABR, data were computed with the Brainstem Toolbox Version 2013 [54]. The FFR was defined as the response following the onset. The fundamental frequency: F0 = 103–121 Hz; the first formant: F1 = 220–720 Hz; and the high frequencies: HF = 756–1130 Hz, were selected and analyzed as part of the FFR. The 11.5 to 46.5 msec portion of the response’s activation magnitude was averaged and divided by the pre-stimulus activation magnitude to populate the root mean square [55]. 

##### Neural Response and Stimulus to Response Consistencies

Inter-response consistency reflects the fidelity of the response morphology when compared with itself [23,55,56]. In this study, the correlation of the responses was calculated in two ways: recordings of odd and even events and by comparing the first 2000 to the second 2000 recordings. The stimulus-to-response consistency describes how the response waveform of the speech-ABRs is similar to the stimulation [23,55,56]. The 10 to 40 msec portion of the stimulus was cross-correlated with each response. This portion contains the harmonics of the stimulation. The Pearson’s r-values are reported for descriptive purposes [57]. However, r-values were converted to z^r^-scores with Fisher’s transformation for all statistical computations [57].

### 2.3. Statistical Analysis

Statistical analysis was completed with IBM SPSS Statistics software (Version 27) (SPSS, Inc., Chicago, IL, USA). To explore differences between groups for the behavioural and electrophysiological tests, the non-parametric Mann–Whitney U test was selected conservatively as the number of participants in the groups was limited and varied between groups and tests.

## 3. Results

### 3.1. Behavioural Tests 

The results of the statistical tests comparing the values between the two groups are presented below for the auditory processing and supplementary tests. As part of this retrospective study, some children in the experimental group had missing data in their files. Children in the control group were prospectively tested and missing data for a few children was due to time constraints (see Table 2). 

#### 3.1.1. Auditory Processing Tests

The Mann−Whitney test results indicated that the children in the experimental group had significantly lower performance than children in the control group in eight of the nine behavioural tests (Table 2): the Dichotic Digits test (right ear: U = 11.50, z = −2.30, *p* = 0.019, r = −0.56, left ear: U = 3.00, z = −3.13, *p* = 0.001, r = −0.76), the Staggered Spondaic Word test (RNC: U = 80.50, z = 2.43, *p* = 0.019, r = 0.54; RC: U = 83.00, z = 2.53, *p* = 0.011, r = 0.57; LC: U = 98.00, z = 3.65, *p* = 0.000, r = 0.82; LNC: U = 82.50, z = 2.51, *p* = 0.011, r = 0.56), the Competing Sentences test, in the left ear only (U = 8.00, z = −1.24, *p* = 0.012, r = −0.76), the Random Gap Detection test (U = 100.00, z = 3.87, *p* = 0.000, r = 0.86), the Pitch Pattern Sequence test (right ear: U = 6.00, z = −3.43, *p* = 0.000, r = −0.77; left ear: U = 7.50, z = −3.30, *p* = 0.000, r = −0.74), the BKB Speech in Noise test in the binaural condition (U = 85.00, z = 2.67, *p* = 0.007, r = 0.60) and the Low Pass Filtered Speech test (right ear: U = 10.00, z = −2.51, *p* = 0.011, r = −0.61; left ear: U = 5.00, z = −3.01, *p* = 0.002, r = −0.73). The effect size for significant results was found to be of a moderate to large effect. 

#### 3.1.2. Supplementary Tests 

The same pattern was observed for the supplementary tasks. The Mann−Whitney test results indicated that the experimental group performed significantly worse than the control group for the Phonemic Synthesis test (U = 11.50, z = −2.56, *p* = 0.009, r = −0.60) and for three of the four conditions of the Auditory Continuous Performance test: Impulsivity (U = 68.00, z = 2.52, *p* = 0.012, r = 0.59), Inattention (U = 73.50, z = 2.99, *p* = 0.001, r = 0.70) and Total (U = 76.00, z = 3.21, *p* = 0.001, r = 0.76). The vigilance component of the test was not significantly different between groups (Table 2). 

### 3.2. Electrophysiological Tests

Electrophysiological data was compiled for the 20 participants. Transient and sustained responses of the peak latency and amplitude were collected and are presented in Table 3 and Figure 1. The response magnitude with root mean square, fundamental frequency and first formant amplitude, and correlations between responses as well as between response and stimulus are reported in Table 4, and Figure 2 and Figure 3.

#### 3.2.1. Transient and Sustained Responses

For the click-ABR, there was a significant difference between groups only for wave I amplitude, which was smaller for the experimental group than for the control group (U = 20.50, z = −2.41, *p* = 0.023, r = −0.50) (Figure 1). The effect size for this analysis (r = −0.50) was found to equate Cohen’s (1988) convention [58] for a large effect (r = 0.50) (Table 3). For the speech-ABR, the experimental group had significantly longer latency than the control group for wave D (U = 87.50, z = 2.843, *p* = 0.003, r = 0.64) and wave E (U = 76.50, z = 2.006, *p* = 0.043, r = 0.45) (Figure 1). The effect size for wave D latency (r = 0.64) and wave E latency (r = 0.45) was found to respectively exceed Cohen’s (1988) convention for a large effect (r = 0.50) and a moderate effect (r = 0.30). The latency of waves V and O was longer for the experimental group than in the control group. Statistically, the test results were close for wave V and wave O, but did not reach a level of significance (U = 72.50, z = 1.712, *p* = 0.089, r = 0.38) and (U = 74.50, z = 1.867, *p* = 0.063, r = 0.42), respectively (Table 3).

Regarding the statistical test results for the root mean square, F0, F1, and HF indicated no significant differences between groups for any amplitude measures despite the fact that the overall amplitude of the component was lower in the experimental group than the control group (Table 4).

#### 3.2.2. Neural Response and Stimulus to Response Consistencies

The Mann−Whitney test results revealed no significant difference between the two groups for neural response consistency, but the odd/even condition approached significance, where the experimental group had lower neural response consistency than the control group (U = 27.00, z = −1.739, *p* = 0.089, r = −0.39) (Figure 2). For the stimulus-to-response consistency (Figure 3), there was no significant group effect (U = 33.00, z = −1.285, *p* = 0.218, r = −0.29).

## 4. Discussion

One objective of the current study was to investigate if LiD in children having academic difficulties could be solely attributed to APD. All children in the experimental group fulfilled the APD diagnostic criteria of ASHA (2005) [4]. They had significantly lower performance on at least two behavioural tests assessing different auditory capacities at two standard deviations compared to the control group. Indeed, they had abnormal results for binaural integration, binaural separation, figure-ground segregation, auditory closure, auditory sequential organization, and temporal resolution abilities. They also had poorer performance than the control group on the auditory sustained attention screening test (ACPT) as well as on the phonemic synthesis test. These results do not support the conclusion that the listening difficulties of the children in the experimental group were specifically caused by APD. Poor performance in the auditory processing test battery has been reported in children with learning disabilities [59,60,61], SLI [62], dyslexia [63], and ADHD [64]. 

Another objective of the study was to explore if click-evoked ABR and FFR were sensitive enough tools to identify children with LiD, and if there was any difference in the neural consistency of FFR responses between children with LiD and TD school-aged children. In click-ABR assessments, there were no significant differences in wave I, III, and V latency; however, the amplitude of wave I was significantly smaller in children with LiD than in the control group. The reduction in wave I amplitude was previously reported in children with APD and language disorders [65,66], and could be attributed to weaker neural synchrony in response to the auditory stimulus at the level of the cochlea in children with LiD [67]. It should also be mentioned that the ABR amplitude could be more variable than latency measures [68]. Considering this point and the sample size, the results should be interpreted with caution. 

Regarding the speech-ABR test results, the latencies of waves V, D, E, and O were longer in children with LiD compared to TD children. The difference was, however, significant only for waves D and E. These findings were in accordance with studies reporting prolonged latencies of waves evoked in response to transient and/or sustained elements of speech stimuli in children with APD, LD, dyslexia, and ADHD [32,33,34,35,36,38,69,70,71]. Atypical encoding of consonants and vowels at the level of the brainstem may lead to poorer speech perception abilities [70]. 

Another finding of this study was the observation of greater variability/inconsistency of neural responses to speech sounds across trials in children with LiD compared to the controls, despite the fact that there was no significant difference in neural consistency between the two groups. Poor response consistency indicates differences in FFR morphology between trials [72]. This inconsistency in speech sound representation could prevent the auditory system from improving the sharpness of neural representation by sound repetition [73]. This can adversely affect the abilities of speech perception in noise [73]. Inter-individual differences were also seen in the stimulus-to-response relationship, where the variability in the responses was greater in children with LiD than the control group. The poor relationship between stimulus and response represents reduced morphology and might indicate poor neural fidelity due to loss of neural synchrony [74].

## 5. Conclusions

Children with LiD and children with typical development were assessed with auditory processing tests, attention and phonemic synthesis screening tests, and click- and speech-ABR measures. The findings revealed poor auditory processing, language, and attentional skills in children in the experimental group, which suggested that their LiD was not related exclusively to APD. The prolonged latency of waves D and E could indicate slower processing of speech stimuli at the brainstem level and might be an indication of an auditory processing deficit. As such, FFR could be used as a complementary objective tool in assessing auditory processing information at the brainstem level in children with LiD and learning problems. However, the low sample size made it difficult to generalize the results of the present study to the population of children with LiD. Further studies with greater sample sizes are needed to determine probable relationships between the results of behavioural tests and specific manifestations of FFR. In the present study, correlation analyses could not be performed due to the limited number of participants. Moreover, studies could also explore whether FFR recordings in adverse listening situations, such as in background noise, can shed further light on neural inconsistencies in processing speech stimuli at the subcortical level in children with LiD. The outcomes of these studies may reveal that the use of FFR measurements in noise and behavioral auditory tests is a suitable combination to identify APD in school-aged children with LiD.

## Figures and Tables

**Figure 1 jcm-12-00897-f001:**
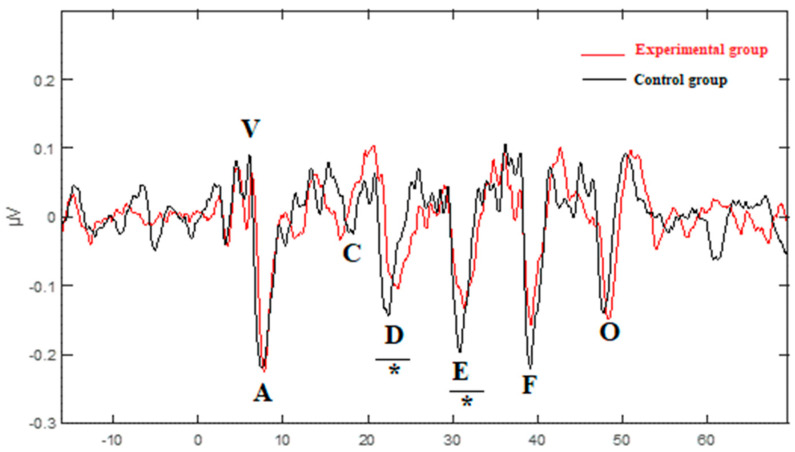
Grand average of speech-evoked ABR in the experimental (red) and control (black) groups. * Significantly different between groups (*p* < 0.05).

**Figure 2 jcm-12-00897-f002:**
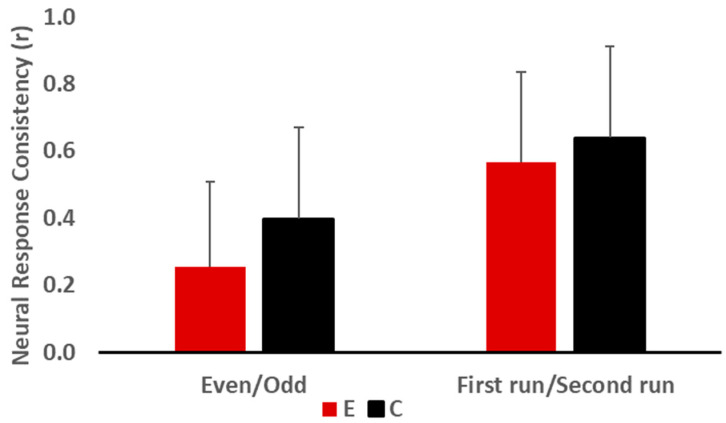
Neural response consistency (r) for the experimental (E) and the control (C) groups for even and odd events and the first and second runs of the recording.

**Figure 3 jcm-12-00897-f003:**
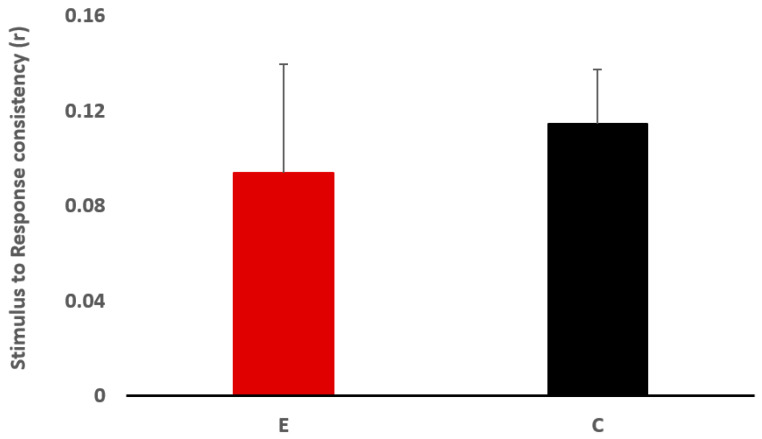
Stimulus to Response consistency (r) for the experimental (E) and the control (C) groups.

**Table 1 jcm-12-00897-t001:** Profile of the ten children in the experimental group.

Participant	Sex	Age (Years: Months)	Reason for Referral/Symptoms	Diagnoses Prior to the Audiological Evaluation	Diagnosed by	Medication
**EXP01**	M	7:11	Indicators of an auditory processing problem such as receptive language struggles, requiring repetitions, with differences between verbal and non-verbal abilities	ADHD, Learning disorder	Psychologist	no meds
**EXP02**	F	9:4	Language disorder	Mixed Expressive-Receptive Language Disorder, ADHD, Specific Language Disorder	Psychologist	no meds
**EXP03**	M	7:0	Following instructions are difficult, and parts may be forgotten before the child can follow through. Struggling with learning to read.	Learning Disability, ADHD	Psychologist	Adderall
**EXP04**	M	7:4	Difficulties in the school environment related to problems hearing in noise experiencing challenges with overall motor skills.	Oculomotor dysfunction Accommodative dysfunction	Optometrist	no meds
**EXP05**	M	7:11	Needed additional time to respond to questions, experienced speech sound confusion, difficulty manipulating the sounds within words, and showed auditory memory weakness.	Communication disorder-expressive language, Learning disorder (Written Expression)	Psychologist	no meds
**EXP06**	M	7:1	Weak phonological awareness and phonological memory, difficulty repeating nonsense words, shorter working memory for verbal information, reduced understanding of spoken information, and weak reading comprehension.	None	Not applicable	no meds
**EXP07**	M	8:4	Phonological awareness, auditory memory, distinguishing sounds in noise and auditory processing difficulties. Auditory attention and sustaining attention were of concern as well.	ADHD, Learning Disorder in Reading and Written expression	Psychologist	Biphentin
**EXP08**	F	10:5	Having trouble with reading comprehension and with decoding words phonetically more so than reading sight words.	None	Not applicable	no meds
**EXP09**	F	8:2	Following a hearing test. Reading has been reported as a challenge for the child.	None	Not applicable	no meds
**EXP10**	M	11:1	Missing instructions from the teacher, needing confirmation for what was heard, struggling with reading comprehension and has difficulty with writing activities for any subjects, with spelling challenges.	Anxiety, depression non-verbal learning disability	Psychologist	no meds

**Table 2 jcm-12-00897-t002:** Mean, standard deviation (SD) and the number of children (*n*) for both groups, as well as the Mann–Whitney U (*p*) and the effect size (Cohen’s r) between the groups for the auditory behavioural tests.

Test	DD		CS		PPST		BKB-SIN			RGDT	
	RE	LE	RE	LE	RE	LE	RE	LE	BIN		
**Experimental group**											
Mean	73.71	54	80.89	32.57	22.7	22.3	4.25	4.81	3.65	36.83	
Median	70	45	88	20	0	0	3.25	3.75	3.25	40	
SD	15.12	22.06	21.66	29.22	37.52	34.09	2.45	3.64	2.11	5.13	
*n*	7	7	9	7	10	10	8	8	10	10	
**Control group**											
Mean	91.75	88.25	89.4	71.78	91.2	88.9	2.15	2.25	1.5	10.72	
Median	92.5	91.25	90	81	96.5	98	2.25	2	1	11.25	
SD	6.67	8.58	4.65	19.31	11.39	13.80	1.42	0.83	0.97	5.61	
*n*	10	10	10	9	10	10	10	10	10	10	
*p*	0.019 *	0.001 *	0.243	0.012 *	0.000 *	0.000 *	0.068	0.068	0.007 *	0.000 *	
Cohen’s r	−0.56	−0.76	−0.28	−0.62	−0.77	−0.74	0.44	0.45	0.60	0.86	
							**Supplementary Tasks**				
**Test**	**SSW**				**LPFST**		**PST**	**ACPT**			
	**RNC**	**RC**	**LC**	**LNC**	**RE**	**LE**		**IN**	**IM**	**Total**	**Vig**
**Experimental group**											
Mean	2.5	10.2	18.6	4.4	62.25	43.38	16.25	14.5	9.63	24.13	1.71
Median	2.5	9.5	21	4	60	40.5	16	16	10.5	27.5	2
SD	1.90	5.18	6.93	4.53	16.52	18.94	4.56	5.83	5.83	9.11	1.11
*n*	10	10	10	10	8	8	8	8	8	8	7
**Control group**											
Mean	0.6	3.2	3.9	0.8	84.00	78.80	77.33	21.9	4.2	1.9	6.1
Median	0	3	4	1	84	76	22	4	2	5.5	1
SD	1.08	1.03	2.56	0.63	9.80	9.25	8.49	1.52	3.85	2.23	5.32
*n*	10	10	10	10	9	9	10	10	10	10	10
*p*	0.019 *	0.011 *	0.000 *	0.011 *	0.011 *	0.002 *	0.009 *	0.001 *	0.012 *	0.001 *	0.536
Cohen’s r	0.54	0.57	0.82	0.56	−0.61	−0.73	−0.60	0.70	0.59	0.76	0.17

* Significant *p* < 0.05. DD = Dichotic Digits test; CS = Competing Sentences test; PPST = Pitch Pattern Sequence test; BKB-SIN = Bamford–Kowal–Bench Speech in Noise test; RGDT = Random Gap Detection test; SSW = Staggered Spondaic Word test; LPFST = Low Pass Filtered Speech test; PST = Phonemic Synthesis test; ACPT = Auditory Continuous Performance test.

**Table 3 jcm-12-00897-t003:** Latency and amplitude of click- and speech-ABR collected in children of the experimental and control groups.

	Latency (ms)			Amplitude (uV)		
	*n*	Mean	SD	*n*	Mean	SD
**Control group**						
*Click−ABR*						
Peak I	10	2.20	0.16	10	0.14 *	0.04
Peak III	10	4.33	0.37	10	0.11	0.07
Peak V	10	6.02	0.35	10	0.18	0.06
*Speech−ABR*						
Peak V	10	6.35	0.27	10	0.13	0.10
Peak A	10	7.38	0.50	10	−0.28	0.07
Peak C	10	18.22	0.50	10	−0.08	0.06
Peak D	10	22.36 *	0.39	10	−0.19	0.08
Peak E	10	30.74 *	0.56	10	−0.26	0.12
Peak F	10	39.11	0.33	10	−0.28	0.12
Peak O	10	47.97	0.41	10	−0.18	0.12
VA complex	10	1.03	0.26	10	0.41	0.13
VA complex area (μV × ms)	10			10	0.21	0.10
VA complex slope (μV/ms)	10	−0.40	0.12	10		
**Experimental group**						
*Click−ABR*						
Peak I	10	2.33	0.40	10	0.08 *	0.07
Peak III	9	4.26	0.49	9	0.12	0.10
Peak V	10	6.12	0.33	10	0.19	0.05
*Speech−ABR*						
Peak V	10	6.47	0.73	10	0.13	0.07
Peak A	10	7.53	0.62	10	−0.27	0.06
Peak C	9	18.62	1.11	9	−0.06	0.04
Peak D	10	23.88 *	1.31	10	−0.19	0.09
Peak E	10	31.52 *	0.80	10	−0.21	0.08
Peak F	10	39.57	0.96	10	−0.24	0.09
Peak O	10	48.40	0.56	10	−0.20	0.05
VA complex	10	1.06	0.21	10	0.39	0.09
VA complex area (μV × ms)	10			10	0.21	0.06
VA complex slope (μV/ms)	10	−0.39	0.12	10		

* Significantly different between groups (*p* < 0.05).

**Table 4 jcm-12-00897-t004:** Root mean square, F0, F1, and HF amplitudes.

	Mean	SD
**Control group**		
F0 amplitude	12.907	4.384
F1 amplitude	3.340	0.675
HF amplitude	1.059	0.293
RMS amplitude	2.807	1.472
**Experimental group**		
F0 amplitude	12.168	7.013
F1 amplitude	2.999	0.762
HF amplitude	0.849	0.212
RMS amplitude	2.205	1.143

## Data Availability

The data can be made available under an official request.

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
