# Peer review of "Assessing Auditory Processing in Children with Listening Difficulties: A Pilot Study"

_jcm, 2023, doi:10.3390/jcm12030897_

Round 1

Reviewer 1 Report

The study is good, however, it suffers from the limited number of the experimental group. I think the authors should mention this in the conclusion section and recommend further study with an increasing number of cases.

In the introduction section: the definition of APD is not appropriate. Please refer to ASHA Definition.  The rationale of the work is not clear. The methodology section is good and comprehensive. However, the authors need to justify the conduction of electrophysiological evaluation in the right ear only.

Results: there is no need to repeat data from table 2 in the text. The table is enough. 

Discussion: is short and is not focusing on the explanation of the results found in the experimental group.

Recommendations and limitations should be added in the conclusion section of this study

Author Response

Thank you for your valuable comments. Please find the responses to the them:

Comment: The study is good, however, it suffers from the limited number of the experimental group. I think the authors should mention this in the conclusion section and recommend further study with an increasing number of cases.

Response: The following information was provided in the conclusion section: However, the low sample size made it difficult to generalize the results of the present study to the population of children with LiD. Further studies with greater sample sizes are needed to determine probable relationships between the results of behavioral tests and specific manifestations of FFR. In the present study, correlation analyzes could not be performed due to the limited number of participants.

Introduction

Comment: In the introduction section, the definition of APD is not appropriate. Please refer to ASHA Definition. 

Response: For many years, ASHA (2005) definition was used in several publications to describe this disorder. However, since 2005, other professional organizations in the audiology field offer their own definition of APD. In this context, we did not want to limit the reference to a single definition in order to inform readers of the existence of the other definitions. From this perspective, we have rather opted to present the common point of these definitions to describe APD. This is the reason that the following sentence was written in the introduction: Many APD definitions have been proposed [4-8] and the majority agree that APDs are failures of the central auditory system that lead to difficulties in processing auditory information.

Comment: The rationale of the work is not clear.

Response: We have added the following paragraph at the end of the introduction, before the objective of the study, in order to help better understand the purpose of the study: In summary, there is little literature on the auditory capacities and function of children with LiD in order to determine whether their difficulties might be caused primarily by APD. Moreover, none of the above studies on FFR investigated the neural consistency of recorded electrophysiological responses in children with LiD.

Methods

Comment: The methodology section is good and comprehensive. However, the authors need to justify the conduction of electrophysiological evaluation in the right ear only.

Response: The justification was added to the section: They were presented at 80 dB SPL in the right ear only as there is evidence of a speech-specific right ear advantage [52] through an insert earphone (EARLINK 3B,3M Auditory Systems, Indianapolis, IN, USA).

Results

Comment: There is no need to repeat data from table 2 in the text. The table is enough. 

Response: The information in the text and Table 2 are complimentary. In the text, the Mann-Whitney values, as well as the z scores for the statistical analysis, are presented. In Table 2, there is the mean, the median, and the standard deviation for both groups. The p and the effect size are repeated, but statistics in the text were presented according to APA requirements, hence the repetition.

Discussion

Comment: Discussion is short and is not focusing on the explanation of the results found in the experimental group.

Response: At the beginning of the Discussion section, we added this sentence in order to provide more information on the results of the experimental group on the behavioral auditory tests: They had significantly lower performance on at least two behavioral tests assessing different auditory capacities at two standard deviations compared to the control group. Indeed, they had abnormal results for binaural integration, binaural separation, figure-ground segregation, auditory closure, auditory sequential organization, and temporal resolution abilities.

Conclusion

Comment: Recommendations and limitations should be added in the conclusion section of this study

Response: It was done as mentioned in the above first comment.

Reviewer 2 Report

The manuscript by Omidvar et al., entitled “Assessing auditory processing in children with listening difficulties: a pilot study” reports a comparison of auditory response between a group of affected children and normal controls in quantitative terms, by employing electrophysiological measures with click and speech auditory brainstem responses (ABR).On the basis of results presented in the manuscript, the authors suggest that a combination of behavioral testing with click ABR and speech ABR can highlight functional and neurophysiological deficiencies in children exhibiting learning and listening issues, especially at the brainstem level.

 The findings reported in the manuscript will be of interest to the readers of this journal.

 The following points may be included:

1.    Do the authors plan to conduct this study on a large scale in the future and/or on a different populations?

2.   Application of this study to provide these tests at a primary school level may be highlighted.

Author Response

Thank you for your valuable comments. Please find the responses to them.

Comment: Do the authors plan to conduct this study on a large scale in the future and/or on a different populations?

Response: The following information was provided in the conclusion section: However, the low sample size made it difficult to generalize the results of the present study to the population of children with LiD. Further studies with greater sample sizes are needed to determine probable relationships between the results of behavioral tests and specific manifestations of FFR. In the present study, correlation analyzes could not be performed due to the limited number of participants.

Comment: Application of this study to provide these tests at a primary school level may be highlighted.

Response:  The following information was added in the conclusion section: The outcomes of these studies may reveal that the use of FFR measurements in noise and behavioral auditory tests is a suitable combination to identify APD in school-aged children with LiD.

Reviewer 3 Report

Interesting work, the topic is very timely. Problems affecting children is a difficult topic and there is still a need to update the available methods and forms of rehabilitation. The structure of the work is appropriate, good introduction to the topic, explanation of definitions, references to current literature. Correct presentation of the material and research methods. Interesting discussion and conclusion. The present study was approved by the Office of Research Ethics and Integrity at the 111 University of Ottawa.

Author Response

We really appreciate your positive feedback on our paper.